# Epidemiology and Clinical Insights of Catheter-Related Candidemia in Non-ICU Patients with Vascular Access Devices

**DOI:** 10.3390/microorganisms12081597

**Published:** 2024-08-06

**Authors:** Giovanni Scaglione, Marta Colaneri, Martina Offer, Lucia Galli, Fabio Borgonovo, Camilla Genovese, Rebecca Fattore, Monica Schiavini, Alba Taino, Maria Calloni, Francesco Casella, Antonio Gidaro, Federico Fassio, Valentina Breschi, Jessica Leoni, Chiara Cogliati, Andrea Gori, Antonella Foschi

**Affiliations:** 1Division of Infectious Diseases, Luigi Sacco Hospital, University of Milan, 20157 Milan, Italy; scaglione.giovanni@asst-fbf-sacco.it (G.S.); galli.lucia@asst-fbf-sacco.it (L.G.); borgonovo.fabio@asst-fbf-sacco.it (F.B.); genovese.camilla@asst-fbf-sacco.it (C.G.); fattore.rebecca@asst-fbf-sacco.it (R.F.); monica.schiavini@asst-fbf-sacco.it (M.S.); andrea.gori@unimi.it (A.G.); foschi.antonella@asst-fbf-sacco.it (A.F.); 2Division of Infectious Diseases, Fondazione IRCCS Policlinico San Matteo, 27100 Pavia, Italy; marta.colaneri@unimi.it; 3Department of Biomedical and Clinical Sciences, University of Milan, 20157 Milan, Italy; martina.offer@unimi.it; 4Division of Internal Medicine, Luigi Sacco Hospital, University of Milan, 20157 Milan, Italy; taino.alba@asst-fbf-sacco.it (A.T.); maria.calloni@asst-fbf-sacco.it (M.C.); francesco.casella@asst-fbf-sacco.it (F.C.); chiara.cogliati@asst-fbf-sacco.it (C.C.); 5Department of Public Health, Experimental and Forensic Medicine, Section of Biostatistics and Clinical Epidemiology, University of Pavia, 27100 Pavia, Italy; federico.fassio01@universitadipavia.it; 6Department of Electrical Engineering, Eindhoven University of Technology, 5600 MB Eindhoven, The Netherlands; v.breschi@tue.nl (V.B.); jessica.leoni@polimi.it (J.L.)

**Keywords:** candidemia, catheter-related bloodstream infection (CRBSI), catheter-associated bloodstream infection (CABSIs), central line-associated bloodstream infection (CLABSI), peripherally inserted central catheter (PICC), midline, femorally inserted central catheter (FICC), centrally inserted central catheter (CICC)

## Abstract

Introduction: Vascular access devices (VADs), namely peripheral VADs (PVADs) and central venous VADs (CVADs), are crucial in both intensive care unit (ICU) and non-ICU settings. However, VAD placement carries risks, notably catheter-related bloodstream infections (CRBSIs). *Candida* spp. is a common pathogen in CRBSIs, yet its clinical and microbiological characteristics, especially in non-ICU settings, are underexplored. Methods: We conducted a monocentric, retrospective observational study at Luigi Sacco Hospital from 1 May 2021 to 1 September 2023. We reviewed medical records of non-ICU adult patients with CVADs and PVADs. Data on demographics, clinical and laboratory results, VAD placement, and CRBSI occurrences were collected. Statistical analysis compared *Candida* spp. CRBSI and bacterial CRBSI groups. Results: Out of 1802 VAD placements in 1518 patients, 54 cases of CRBSI were identified, and *Candida* spp. was isolated in 30.9% of episodes. The prevalence of CRBSI was 3.05%, with *Candida* spp. accounting for 0.94%. Incidence rates were 2.35 per 1000 catheter days for CRBSI, with *Candida albicans* and *Candida* non-albicans at 0.47 and 0.26 per 1000 catheter days, respectively—patients with *Candida* spp. CRBSI had more frequent SARS-CoV-2 infection, COVID-19 pneumonia, and hypoalbuminemia. Conclusions: During the COVID-19 pandemic, *Candida* spp. was a notable cause of CRBSIs in our center, underscoring the importance of considering *Candida* spp. in suspected CRBSI cases, including those in non-ICU settings and in those with PVADs.

## 1. Introduction

Vascular access devices (VADs) are widely used in intensive care unit (ICU) and non-ICU wards for hydration, blood sampling, supportive therapies, and the infusion of medications and solutions compatible with the chosen venous route. Peripheral VADs (PVADs) are the standard care for non-ICU patients and include short peripheral cannulas (less than 6 cm), long peripheral cannulas or “mini-midlines” (6–15 cm), and midline catheters (20–25 cm). Central venous VADs (CVADs) are selected for patients with complex needs, such as those that require multiple infusions, administration of vasopressors, chemotherapeutic drugs, hypertonic solutions, or hemodynamic monitoring [1,2,3]. These CVADs can be peripherally inserted central catheters (PICCs), femorally inserted central catheters (FICCs), or centrally inserted central catheters (CICCs); however, the peripheral and femoral exit sites are becoming increasingly popular, particularly in non-ICU settings, and have recently become the standard of care due to the security of the insertion and the absence of pneumothorax risk [4]. The placement of VADs is not without risks. Complications might be local, such as phlebitis, exit-site infections, and tissue damage, or systemic, like venous thrombosis and bloodstream infections (BSIs) [5].

Data on BSI episodes in patients with a VAD can be reported using either surveillance terms, which are not established diagnostic criteria, or using the catheter-related BSI (CRBSI) definition, which confirms the catheter as the source of infection [6].

CRBSI is diagnosed when the same organism is isolated from both a blood culture and the tip culture and the quantity of organisms isolated from the tip is greater than 15 colony-forming units (CFUs). Alternatively, differential time to positivity (DTP) requires isolating the same organism from a peripheral vein and a catheter lumen blood culture, with growth detected 2 h earlier (i.e., 2 h less incubation) in the sample drawn from the catheter. CRBSIs are particularly concerning since they are typically associated with increased morbidity, mortality, and prolonged hospital stays [6].

The diagnosis of CRBSI requires microbiological evidence, with *Coagulase-negative Staphylococci* (CoNS), *Staphylococcus aureus* (*S. aureus*), Gram-negative bacteria, and *Candida* species (spp.) being the most frequently isolated pathogens [7]. Nosocomial CRBSIs carry a high mortality risk. *Candida* spp., in particular, has been identified as an independent mortality risk factor. Therefore, prompt catheter removal and the initiation of appropriate antifungal therapy are essential [8].

Patients who did not undergo complete microbiological examinations required by the CRBSI definition were not classified as having CRBSIs, even if the BSI episode was clearly related to the catheter; instead, they were categorized using surveillance terms.

Numerous studies on CRBSIs involving CVADs have been published in recent years, particularly for those caused by *S. aureus* and CoNS, as well as for ICU-acquired infections, where FICCs and CICCs are primarily employed [9,10]. Conversely, there is a limited but growing body of evidence about the infectious risks associated with midline catheters and PICCs [11]. Moreover, while *Candida* spp. remains a leading cause of CRBSIs, the clinical and microbiological characteristics and specific risk factors of *Candida*-related CRBSIs have seldom been explored, particularly in non-ICU settings [12].

The main goal of this study is to describe the epidemiology of *Candida* spp. CRBSI and these infections’ clinical and microbiological features in hospitalized non-ICU patients undergoing CVAD and PVAD placement.

This risk information about *Candida* CRBSI could aid clinicians in identifying at-risk patients who may benefit from empirical antifungal therapies in non-intensive care settings. Additionally, we hope this study will generate increased interest in this topic, encouraging more research on *Candida* spp. CRBSIs beyond the typical patient groups, such as those in intensive care units or home parenteral nutrition programs.

## 2. Materials and Methods

### 2.1. Study Design and Clinical Setting

This monocentric, retrospective, observational study was conducted at Luigi Sacco Hospital from 1 May 2021 to 1 September 2023, after a vascular access team (VAT) with standardized procedures and data reporting formats was established in 2018. All the devices were positioned following the protocol “Safe insertion of PICCs (SIP)” [13].

### 2.2. Study Population

Medical records of hospitalized non-ICU adult patients undergoing CVAD (PICC, FICC, CICC) and PVAD (mini-midline, midline) placement were retrospectively retrieved.

Inclusion criteria were hospitalized adults over 18 years old who had CVAD or PVAD placement. Meanwhile, the exclusion criteria were

Patients who received only short peripheral catheters.Those who received both VAD placement and CRBSI diagnosis in the ICU or within 48 h of being transferred to a non-ICU department.

Furthermore, for patients with CRBSI, only the first episode was included in the analyses.

### 2.3. Ethics

All subjects gave their written informed consent for inclusion. The study was conducted according to the Declaration of Helsinki, and the protocol was approved by the Luigi Sacco Hospital Institutional Review Board (Research Ethics Committee approval number 2021/ST/180).

### 2.4. Definitions

Confirmed deep-seated candidiasis consisted of a specialist-driven diagnosis of *Candida* spp. invasion of the eye, lungs, heart, gastrointestinal tract, central nervous system, or other parenchymal organs, and at least one positive blood culture for *Candida* spp., according to international guidelines [14].

Suspected deep-seated candidiasis consisted of candidemia combined with clinical suspicion of involvement of any of the body sites mentioned above, for which appropriate diagnostic techniques were not performed or provided inconclusive results in a patient with suggestive clinical features (e.g., candidemia and critical post-surgical intra-abdominal infection nonresponding to maximal antibiotic therapy).

The terms CABSI and CLABSI were defined as indicated in the 2024 Infusion Nursing Society Standards of Practice and by the Centers for Disease Control and Prevention’s National Healthcare Safety Network [6]. CABSIs refer to BSIs originating from either PVADs and/or CVADs unrelated to an infection at another site. The CLABSI refers to BSI originating from CVADs unrelated to an infection at another site. Both definitions are not established diagnostic criteria because they refer to a primary BSI in a patient who had a VAD the day of or the day before infection and had a vascular access device for more than two days.

CRBSI was defined with the presence of the following criteria:Differential time to positivity (DTP): The same microorganism is isolated in blood cultures drawn from a peripheral vein and the VAD, which were taken simultaneously. VAD cultures become positive at least two hours before peripheral vein cultures.Isolation of the same microorganism from the catheter’s tip and blood cultures drawn from a peripheral vein [15].

### 2.5. Available Data

Patients’ electronic records comprised demographics (age, sex), clinical data (past medical history, comorbidities, treatments, outcomes), and laboratory results. The latter included biochemistry (albuminemia) and microbiology data such as SARS-CoV-2 tests, blood cultures, urine cultures, colonization surveillance swabs, and susceptibility testing with minimum inhibitory concentration (MIC) values for first-line antimicrobial and antifungal agents, according to international standards [16]. Furthermore, VAD placement data were obtained from the standardized data collection sheet, which included both catheter (type, number of lumens) and placement (body site, date of placement and removal, intra-procedural complications) data, as well as the occurrence of CABSI/CLABSI and CRBSI, and the reason for catheter removal.

### 2.6. Outcomes

The study’s primary outcome was determining the incidence and prevalence of *Candida albicans* and non-albicans *Candida* spp. CRBSI in non-ICU hospitalized patients.

The secondary outcomes were

(1)To describe the clinical and microbiological characteristics of *Candida* spp. CRBSI patients.(2)To compare the clinical characteristics of patients with *Candida*-related CRBSI vs. bacterial CRBSI.

### 2.7. Statistical Analysis

Continuous variables are expressed as medians (interquartile range [IQR]) and categorical variables as counts and percentages. Differences between CRBSI groups (*Candida* spp. Vs. bacterial etiology) were tested with the Mann–Whitney test for continuous variables and Fisher’s exact test or the Chi-square test, as appropriate, for categorical variables. Data were analyzed using R software, version 4.3.3, and statistical significance was accepted at the 5% level.

## 3. Results

### 3.1. General Population and VAD Data

During the study period, 1802 VAD placements were performed on 1518 patients. Clinical and microbiological data for these cases were obtained from electronic medical records, which revealed that CRBSI complicated 54 placements.

The study population consisted of a nearly equal number of males and females, with 52.7% of the participants being female. The median age was 78 years (IQR: 64, 85) (Table 1).

Three hundred thirty-seven patients (22.2%) infected with SARS-CoV-2 underwent VAD placement, of which 70.3% had COVID-19-related pneumonia. Additionally, 20% of the patients were colonized by multidrug-resistant pathogens, and 3.7% had a history of previous CRBSI.

Midline catheters were the most frequently placed VADs, accounting for 1233 placements (68.4%); 553 had mid-thigh exit sites and 680 had upper-limb exit sites. PICCs (401, 22.3%) came in second, with other central venous catheters (CICCs and FICCs) accounting for 9.3% of instances. These catheters were used for more than two days of parenteral nutrition in 322 patients (21.3%).

*Candida* spp. was involved in 17 venous catheter-related infections (30.9%) and 5 co-infections with other bacteria (Figure 1) (Table 2).

Nine *Candida* spp. CRBSIs occurred in patients with midlines out of 1233 midline placements (0.73%); there were six in PICC carriers (6/401, 1.5%), and two patients had other CVADs. Among the 54 bacterial isolates collected (Figure 2), nearly half (25 isolates, 46.3%) were CoNS, predominantly *Staphylococcus epidermidis*, followed by *Enterococci,* both often resistant to first-line agents (53.8% *Methicillin-resistant S. epidermidis* or MRSE, and 25% *Vancomycin-resistant Enterococci* or VRE rates, respectively).

During the study period, 228 patients (15%) died, including 16 from the CRBSI group, resulting in a mortality rate of 29.6% for this group. Seven of these sixteen deaths (43.8%) were attributed to the CRBSI event.

When looking at the *Candida* spp. CRBSI group (Table 3), 5 out of 17 patients died, and three deaths (60%) were deemed to be closely related to the *Candida* spp. CRBSI event.

The overall prevalence of CRBSI was 3.05%, while the prevalence of *Candida* spp. CRBSI was 0.94%. Furthermore, the incidence was 2.35 per 1000 catheter days for CRBSI, with specific incidences of 0.47 and 0.26 per 1000 catheter days for CRBSIs caused by *Candida albicans* and non-albicans isolates, respectively (Table 1).

When looking at the different VAD types, *Candida* spp. CRBSI incidence was 0.55 per 1000 catheter days for central venous catheters and 0.17 per 1000 catheter days for midline catheters.

**Table 1 microorganisms-12-01597-t001:** Characteristics of the study population who underwent VAD placement.

	Total
Number of patients	1518
Number of VAD placements	1802
VAD placements per patient	1.18
Age (Q1, Q3)	78 (64, 85)
Male sex (%)	717 (47.23%)
Transferred from the ICU (%)	62 (4.08%)
SARS-CoV-2 infection (%)	337 (22.2%)
COVID-19-related pneumonia (%)	237 (15.61%)
Overall CRBSI prevalence	3.05%
Overall CABSI prevalence	4.5%
*Candida* spp. CRBSI prevalence	17 events (0.94%)
*Candida* spp. CRBSI prevalence in midline catheters	0.22%
*Candida* spp. CRBSI prevalence in central venous catheters	0.72%
Overall CRBSI incidence (per 1000 catheter days)	2.35
Overall CABSI incidence (per 1000 catheter days)	3.46
*Candida albicans* CRBSI incidence (per 1000 catheter days)	0.47
Non-albicans *Candida* CRBSI incidence (per 1000 catheter days)	0.26
*Candida* spp. CRBSI incidence in midline catheters (per 1000 catheter days)	0.17
*Candida* spp. CRBSI incidence in central venous catheters (per 1000 catheter days)	0.55
Median dwell time VAD days (Q1, Q3)	13 (8, 21)
VAD type (%)	
Midline with upper-limb exit site	680 (37.74%)
Midline with mid-thigh exit site	553 (30.68%)
PICC	401 (22.25%)
CICC	72 (3.99%)
FICC	96 (5.33%)
Median albuminemia (g/L)	27 (23, 30)
Parenteral nutrition (%)	322 (21.21%)
MDR colonization (%)	303 (19.96%)
Outcomes	
Transferred to the ICU (%)	45 (2.96%)
Death (%)	228 (15.02%)
CRBSI-related death (%)	17 (7.46%)

History of CABSI or CRBSI consisted of any patient whose medical record included an accurate description of previous catheter-related infections, ICU: Intensive care unit, VAD: Vascular access device, MDR: Multidrug-resistant.

**Figure 1 microorganisms-12-01597-f001:**
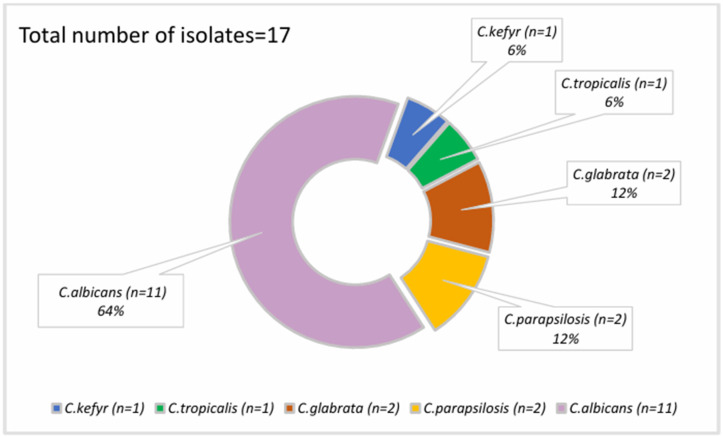
*Candida* spp. isolates in CRBSI episodes.

**Figure 2 microorganisms-12-01597-f002:**
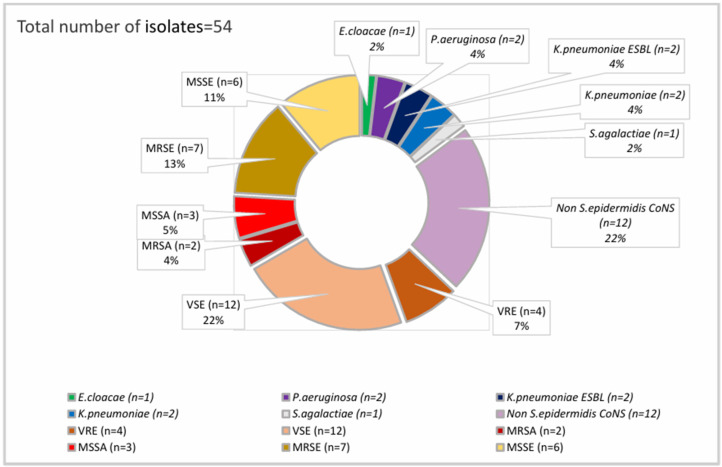
Bacterial isolates in CRBSI episodes. MSSA: Methicillin-sensitive *S. aureus*, MRSA: Methicillin-resistant *S. aureus*, MSSE: Methicillin-sensitive *S. epidermidis*, MRSE: Methicillin-resistant *S. epidermidis*, Enterobacterales WT: *E. Cloacae* Sensitive to III gen. cefalosporins, ESBL: Resistant to III gen. cefalosporins and piperacillin-tazobactam, VSE: Vancomycin-sensitive *Enterococci*, VRE: Vancomycin-resistant *Enterococci*, CoNS: Coagulase-negative *Staphylococci*.

### 3.2. Description of Candida *spp.* CRBSI Events

In the 17 fungal CRBSIs, C. *albicans* was isolated in most cases, accounting for 11 of the Candida spp. CRBSI episodes. Non-albicans *Candida* spp. isolates were retrieved in six cases—two *Candida glabrata* (*C. glabrata*) isolates, two *Candida parapsilosis* (*C. parapsilosis*) isolates, one *Candida kefyr* isolate, and one *Candida tropicalis* isolate (Figure 1).

Median age was 79 (IQR: 64, 88), and more than half of the patients (9, 52.9%) were infected with SARS-CoV-2, five of whom had pneumonia. Two-thirds (64.7%) were colonized by *Candida* spp., though only four had colonization at two different body sites, notably at the urinary tract, airways, and skin. Sixteen patients had a bacterial infection in the preceding 28 days, with 18.8% progressing to septic shock and 37.5% receiving five or more antimicrobial agents before the candidemia diagnosis. Among the known candidemia risk factors, no patients underwent dialysis, but 10 (58.8%) met the KDIGO criteria for acute kidney injury (AKI) with a mean score of 1.6. Additionally, 47.1% of patients had abdominal surgery, while alcohol abuse and gastrointestinal perforation were noted in one patient each (Table 2).

**Table 2 microorganisms-12-01597-t002:** Clinical and microbiological features of patients with *Candida* spp. CRBSI.

	Total Number of Patients = 17
*Candida* spp. colonization	64.71%
Number of colonized sites (Mean ± SD)	0.88 ± 0.76
Previous bacterial infection	94.12%
Previous septic shock	17.65%
*Candida* spp. and bacterial co-infection (%)	29.41%
Median AKI score (Q1, Q3)	1 (0, 2)
Abdominal surgery (%)	47.06%
Alcohol abuse (%)	5.88%
Gastrointestinal perforation (%)	5.88%
Five or more previous antibiotics (%)	35.29%
Non-albicans candidemia (%)	35.29%
Confirmed deep-seated infection (%)	23.5%
Suspected deep-seated infection (%)	23.5%
Caspofungin empirical therapy (%)	84.62%
Fluconazole CLSI S or I (%)	75%
Caspofungin CLSI S or I (%)	100%
VAD removed after CRBSI diagnosis	88.24%

Four individuals had confirmed deep-seated candidiasis, three of whom presented with endophthalmitis. Four more patients were diagnosed with suspected deep-seated candidiasis, all with suspected post-surgical *Candida* infection of the gastrointestinal tract.

Treatment, as well as second- or third-line diagnostic techniques, were not pursued due to a palliative approach in two patients, and there was a rapid deterioration leading to death before obtaining microbiology results in the remaining two cases. Thus, of the 17 patients, only 13 received treatment. Caspofungin was used as a first empiric agent in 11 cases (84.6%). Upon the availability of susceptibility testing, echinocandin treatment was either continued as monotherapy (four patients), switched to fluconazole monotherapy (five patients), or combined with fluconazole in two severe candidemia cases with endophthalmitis. All *Candida* spp. isolates were sensitive to caspofungin, but only 75% were sensitive to fluconazole. Timely VAD removal, according to international guidelines and clinical expertise, was performed in all but two patients since a palliative approach with intravenous supportive therapies had already been started for concurrent comorbidities. Data on susceptibility to first-line antifungal agents for the three most common isolates are depicted in Table 4.

### 3.3. Comparison between Candida *spp.* and Bacterial CRBSI

When comparing the *Candida* spp. and bacterial CRBSI groups (Table 3), SARS-CoV-2 infection, COVID-19-related pneumonia, and hypoalbuminemia were significantly associated with *Candida* spp. CRBSI (*p*-values of 0.002, 0.002, and 0.038, respectively). Longer catheter dwelling times were related to bacterial CRBSI (*p* = 0.005).

**Table 3 microorganisms-12-01597-t003:** Comparison between the patients with *Candida* and bacterial CRBSI events. Followed by statistical significance of the Mann–Whitney test and Fisher’s exact test assessing inter-group differences between bacterial and fungal infections (right column).

	*Candida* spp. CRBSI (N = 17)	Bacterial CRBSI (N = 37)	*p*-Value
Age (Q1, Q3)	79 (64, 88)	73 (60, 81)	0.268
Male sex (%)	8 (47.06%)	16 (43.24%)	1.000
History of CABSI/CRBSI (%)	2 (11.76%)	3 (8.11%)	0.645
Charlson comorbidity Index (Q1, Q3)	6 (4, 7)	6 (4, 7)	0.764
Comorbidity count (Q1, Q3)	3 (2, 4)	3 (2, 3)	0.703
Diabetes (%)	5 (29.41%)	9 (24.32%)	0.745
CVD (%)	7 (41.2%)	15 (40.54%)	1.000
COPD (%)	6 (35.29%)	6 (16.22%)	0.162
Neurological disorder (%)	8 (47.06%)	17 (45.95%)	1.000
Liver cirrhosis (%)	1 (5.88%)	2 (5.41%)	1.000
Immunosuppression	6 (35.29%)	21 (56.76%)	0.241
CKD (%)	2 (11.76%)	3 (8.11%)	0.645
Obesity (%)	1 (5.88%)	1 (2.70%)	0.535
Transferred from the ICU (%)	2 (11.76%)	2 (5.41%)	0.582
SARS-CoV-2 infection (%)	9 (52.94%)	4 (10.8%)	**0.002**
COVID-19-related pneumonia (%)	5 (29.41%)	0 (0%)	**0.002**
Median VAD days (Q1, Q3)	10 (8, 20)	22 (15, 34)	**0.005**
Parenteral nutrition (%)	9 (52.94%)	23 (62.16%)	0.563
VAD type (%)			0.374
Midline	9 (52.94%)	11 (29.73%)	-
PICC	6 (35.29%)	21 (56.76%)	-
CICC	1 (5.88%)	2 (5.41%)	-
FICC	1 (5.88%)	3 (8.11%)	-
Median albuminemia (g/L)	24 (22, 27)	28 (24, 33)	**0.038**
MDR colonization (%)	5 (29.41%)	10 (27.03%)	1.000
Death (%)	5 (29.41%)	10 (27.03%)	1.000
CRBSI-related death (%)	3 (60.0%)	5 (50.0%)	1.000
Transferred to the ICU (%)	1 (5.88%)	2 (5.41%)	1.000

History of CABSI or CRBSI consisted of any patient whose medical record included an accurate description of previous catheter-related infections, CVD: Cardiovascular disease, COPD: Chronic obstructive pulmonary disease, CKD: Chronic kidney disease, ICU: Intensive care unit, VAD: Vascular access device, MDR: Multidrug-resistant.

**Table 4 microorganisms-12-01597-t004:** Susceptibility to first-line antifungal agents of the three most common *Candida* spp. isolates according to Clinical Laboratory and Standards Institute (CLSI) version M60, 2017.

	*C. albicans* CLSI MIC (Number of Isolates = 10)								
	0.03	0.06	0.12	0.25	0.5	1	2	4	>=8
Fluconazole	1	-	1	4	3	1	-	-	-
Caspofungin	2	6	2	-	-	-	-	-	-
	***C. glabrata* CLSI MIC** (Number of isolates = 2)								
	**0.03**	**0.06**	**0.12**	**0.25**	**0.5**	**1**	**2**	**4**	**>=8**
Fluconazole	-	-	-	1	-	-	-	-	1
Caspofungin	2	-	-	-	-	-	-	-	-
	***C. parapsilosis* CLSI MIC** (Number of isolates = 2)								
	**0.03**	**0.06**	**0.12**	**0.25**	**0.5**	**1**	**2**	**4**	**>=8**
Fluconazole	-	-	1		-	1	--	-	-
Caspofungin	-	-	1	--	1	-	-	-	-

## 4. Discussion

In the study population, we identified 17 *Candida* spp. CRBSI episodes; there were six in PICC carriers and nine in midline catheter carriers, with an overall prevalence rate of 0.94%. The incidence rates of *Candida albicans* and non-albicans *Candida* CRBSIs were 0.47 and 0.26 per 1000 catheter days, respectively.

Furthermore, when comparing *Candida* spp. and bacterial CRBSI cases, there were no statistically significant differences in demographics or underlying comorbidities. On the other hand, *Candida* spp. CRBSI was significantly associated with SARS-CoV-2 infection, COVID-19-related pneumonia, lower median albumin levels, and shorter catheter dwell times.

Midlines and PICCs were widely used, with the latter being twice more frequently infected than the former, displaying three-times-higher CRBSI incidence per 1000 catheter days.

Interestingly, no specific comparison is available for *Candida* spp. CRBSI risk, but a two- to four-fold overall increase in catheter-related BSI risk for PICCs over midlines has been recently described [17]. These results are consistent with the different indications for the two VAD types since PICC catheters are preferred to midlines when parenteral nutrition and longer dwell times are expected.

Despite the non-ICU setting and a well-structured VAT, our study population presented CRBSI rates at the upper end of the range defined in the literature. Maki et al.’s extensive review identified CRBSI incidence rates as being the lowest in peripheral VAD carriers (0.2 per 1000 catheter days) and the highest in CVAD carriers (2.7 per 1000 catheter days) [18].

This observation may be attributed to the study’s timeframe, which coincided with the SARS-CoV-2 pandemic, which greatly exacerbated hospital-acquired infections (HAIs) in Northern Italy [19].

Solid evidence supports the increase in CRBSIs and candidemia during pandemics, linking the rise in HAIs to healthcare system overload, prolonged hospital stays, and the increased usage of immunosuppressive treatments [20,21,22]. Nonetheless, this is, to our best knowledge, the first study to report *Candida* spp. CRBSI incidence and prevalence rates in non-ICU settings.

Comparing our data to another multicentric retrospective study performed during the first wave of the COVID-19 pandemic, we can confirm the higher risk of CRBSI in those patients [23].

Furthermore, existing evidence on this topic shows very high mortality rates in COVID-19 patients who then develop candidemia, describing pooled in-hospital mortality rates above 60% in emergency settings in Northern Italy during the first COVID-19 waves [24].

In our study, we describe *Candida* spp. CRBSI mortality rates of almost 30% (29.4%), even though statistical significance was not reached when compared to mortality rates of bacterial CRBSI (27%). The *Candida* spp. superinfection in the COVID-19 subgroup was not analyzed specifically due to limited sample sizes; nonetheless, the numbers remain high.

Lastly, SARS-CoV-2 infection and COVID-19-related pneumonia were associated with a higher likelihood of *Candida* spp. than bacterial CRBSI. This has been described indirectly, but no study directly links SARS-CoV-2 infection or COVID-19 pneumonia to this phenomenon. These considerations stress the importance of correctly managing VADs in critical or immunosuppressed patients in non-ICU settings [25].

Unexpectedly, hypoalbuminemia was more frequently encountered in *Candida* spp. than in bacterial CRBSI. Scarce evidence on this association is currently available, although recent in vitro data endorse it [26]. Hypoalbuminemia is more commonly observed in critical patients, as well as in those who underwent abdominal surgery or have other gastrointestinal (pancreatitis, GI perforation, alcohol abuse, conditions requiring parenteral nutrition) or nephrological (AKI, dialysis) disorders. Accordingly, all these conditions, for which we noted high prevalences in our cohort, have already been associated with candidemia [27].

Patients with *Candida* spp. CRBSI did not have longer catheter indwelling times than those with bacterial etiology CRBSIs. These data would lead us not to consider the hospitalization time and the placement of the VAD as decisive in calculating the risk of infection by *Candida* spp. Furthermore, the isolation of *Candida* spp. in a single blood culture obtained from any catheter warrants prompt removal of the VAD, whereas a more conservative approach, leading to 5 days of lock therapy and retention of VAD, is accepted today for most bacterial isolates when successful [28].

Thus, in our experience, clinical management was complex and followed international guidelines, with almost all patients undergoing timely VAD removal and empiric echinocandin antimycotic treatment. Although non-albicans *Candida* spp. isolates accounted for more than one-third of the total, MIC values were favorable, and no resistance to echinocandins and little resistance to fluconazole were observed (Table 4), in line with the non-intensive care setting of the study.

Some limitations must be declared. Firstly, the study’s retrospective nature may have introduced inherent biases and limitations in data collection and analysis. Secondly, the single-center design makes the study’s findings not fully generalizable to other settings, populations, or healthcare contexts, limiting the external validity.

Moreover, the study’s timeframe included the COVID-19 pandemic, which undoubtedly increased the number of catheter-related infections outside the ICU [23]. However, collecting data throughout the SARS-CoV-2 pandemic hampers the generalizability of the findings, mainly since our hospital was a COVID-19 reference center for the metropolitan area of Milan.

Finally, the limited number of *Candida* spp. infections did not allow for extensive sub-group analyses. On the other hand, the main strength of our study is the in-depth characterization of almost 2000 VAD placements focused on correct identification and comprehensive characterization of *Candida* spp. CRBSI and other concurring infective episodes. All VAD placements were performed by the same VAD team that completed data entry on standardized data collection forms.

To our knowledge, this is the first study to provide *Candida* spp. CRBSI rates in non-ICU settings. Finally, our study is one of the few that assesses *Candida* spp. infections in PVADs or CVADs. These findings add to the current body of evidence demonstrating a substantial risk of CRBSI in non-ICU patients with PICCs and midlines [17,29].

## 5. Conclusions

Our data from a large Northern Italian hospital during the COVID-19 pandemic revealed a high prevalence of CRBSI infections caused by *Candida* spp. We found that COVID-19 and hypoalbuminemia were more frequently associated with *Candida* spp. CRBSI compared to bacterial CRBSI.

These two factors can be interpreted as markers of patient frailty and severity rather than direct infection causes. Thus, in high-risk patients deteriorating under appropriate empirical therapy, *Candida* spp. should be considered as a potential CRBSI cause even in non-ICU wards.

## Data Availability

The complete dataset is available and can be shared upon reasonable request from the corresponding author for privacy and data safety purposes.

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
