# Peer review of "Epidemiology and Clinical Insights of Catheter-Related Candidemia in Non-ICU Patients with Vascular Access Devices"

_microorganisms, 2024, doi:10.3390/microorganisms12081597_

Round 1

Reviewer 1 Report

Comments and Suggestions for Authors

The authors attempt to describe the epidemiology of Catheter related bacteremia in non ICU patients with vascular access devices.

I have several remarks which I include in the uploaded file.

My general remarks are the following. 1. You include in the same group patients with peripheral and central VADs. In my opinion this is not a correct approach.2. You are using a definition of CABSI which includes BSIs originating from peripheral veins. Where is this definition reported? You have no reference. The usual and well known definition includes "A CA-BSI is defined as a BSI with a recognized pathogen that is not related to another infection. If a central line (defined as a vascular access device that terminates at or close to the heart or in one of the great vessels) was in use at any time during the 48 h prior to the onset of the BSI, the BSI is considered CA." No peripheral lines are included in the definition. So I think that patients with midlines should be excluded.

3. The Covid 19 does not appear in the title. It appears in the conclusions and you also talk about Covid pandemic while the data you have are partly during the pandemic

Comments on the Quality of English Language

English language is generally satisfactory.

It needs some corrections, which I have noticed in the uploaded manuscript.

Author Response

Dear Editor,

Enclosed is our revised manuscript entitled “Epidemiology and Clinical Insights of Catheter-related Can-didemia in Non-ICU Patients with Vascular Access Devices.” We want to thank the Editor and the reviewers for their constructive criticism, which substantially improved our paper. We hope that the paper will be suitable for publication. We have made a point-by-point answer to the referees’ comments below.

Answers to Reviewer 1

Reviewer 1 comment: The authors attempt to describe the epidemiology of Catheter related bacteremia in non ICU patients with vascular access devices. I have several remarks which I include in the uploaded file.

Response to Reviewer 1 comment: We thank the reviewer for his/her correction. The uploaded file is not our paper. Its title is “An Update on Viral Hepatitis B and C in Mexico: Advances and 2 Pitfalls in Eradication Strategies.” Could you send us the correct version?

Reviewer 1 comment: My general remarks are the following. 1. You include in the same group patients with peripheral and central VADs. In my opinion this is not a correct approach.

Response to Reviewer 1 comment:

We thank the reviewer for his/ her precious remarks. Our paper, following up on previous work from our study group (available at: https://pubmed.ncbi.nlm.nih.gov/?term=%28crbsi%29+AND+%28gidaro%29&sort=relevance), thoroughly analyzes catheter-related infections in both midline and central venous catheters, and, as stated in the backgrounds and methods section of the study we did not include short peripheral cannulas (less than 6 cm) or long peripheral cannulas or "mini-Midlines" (6-15 cm) in the analyses. Thus, the peripheral VAD used in our paper is a midline catheter (25 centimeters long), with the tip placed inside the chest in the axillary or subclavian vein, sometimes even in the brachiocephalic vein. When looking at the results section (Table 2), we also considered the various VAD types separately. We commented in the discussion on infection rates in peripheral, peripherally inserted, and central vascular devices. Furthermore, based on your considerations, we also calculated Candida spp. CRBSI rates per 1000 catheter days for both Midline and central venous catheters (Table 2) to provide a more precise incidence value related to the VAD type used.

Reviewer 1 comment: 2. You are using a definition of CABSI which includes BSIs originating from peripheral veins. Where is this definition reported? You have no reference. The usual and well-known definition includes "A CA-BSI is defined as a BSI with a recognized pathogen that is not related to another infection. If a central line (defined as a vascular access device that terminates at or close to the heart or in one of the great vessels) was in use at any time during the 48 h prior to the onset of the BSI, the BSI is considered CA." No peripheral lines are included in the definition. So I think that patients with midlines should be excluded.

Response to Reviewer 1 comment:  We appreciate the reviewer’s thoughtful criticism. As declared just in the first version of our paper, the latest definition of CABSI is according to the latest Infusion Nursing Society Standards of Practice published in January 2024, quote number 6 (6. Nickel B, Gorski L, Kleidon T, et al. Infusion Therapy Standards of Practice, 9th Edition. J Infus Nurs. 2024 Jan-Feb 01;47(1S Suppl 1):S1-S285. doi: 10.1097/NAN.0000000000000532. PMID: 38211609.). The definition is reported on page 170 of Nickel et al. paper.

The previous definition provided by the same authors is the exact one we reference and can be found in the previous version of the Infusion Nursing Society Standards of Practice 8th edition, published in 2021, on page 153. (Gorski LA, Hadaway L, Hagle ME, et al. Infusion therapy standards of practice, 8th edition. J Infus Nurs 2021; 44(1S Suppl 1): S1–S224.) This paper was not quoted in our paper for redundancy.

For further references on how we defined CABSI and CRBSI in previous works: https://pubmed.ncbi.nlm.nih.gov/?term=%28crbsi%29+AND+%28gidaro%29&sort=relevance

Despite this, we forgot to insert quote 6 near the CABSI definition; we rectified the mistake in the revised version.

Reviewer 1 comment: 3. The Covid 19 does not appear in the title. It appears in the conclusions and you also talk about Covid pandemic while the data you have are partly during the pandemic

Response to Reviewer 1 comment:  We thank the reviewer for his/her precious feedback. We omitted COVID-19 in the title as this is not the prime focus of our work. Detailed and consistent descriptions of each catheter placement were collected to create a detailed dataset that coincidentally started during the COVID-19 pandemic. This was an unavoidable event, altering measures and estimations and having to be addressed while discussing and commenting on the study results. This issue is addressed in the limitations part of the discussion section. It has now been corrected in the revised manuscript, stressing that the results provided have to be interpreted according to the unique study’s setting and timeframe. Additionally, we clearly stated in the conclusions that COVID-19 was meant to be considered as a marker of patient frailty and medical burden rather than a potential cause of the catheter infection.

On the timeframe: The COVID-19 public health emergency was deemed over by WHO’s Director-General in May 2023, and we enrolled patients up to August 2023, the number of cases was low during all summer months, but those of 2020, and our center, being among the references for this disease, received a steady in-flow of COVID-19 patients throughout the whole study period.

https://www.who.int/director-general/speeches/detail/who-director-general-s-opening-remarks-at-the-media-briefing---5-may-2023

Reviewer 1 comment: English language is generally satisfactory. It needs some corrections, which I have noticed in the uploaded manuscript.

Response to Reviewer 1 comment: We thank the reviewer for his/her correction. Once the updated file will be uploaded, we will address the language corrections adequately.

Answers to Reviewer 2

Reviewer 2 comment: The subject is not new, but it brings more details about the association between fungal CRBSI and some data of the patients, such as their immune status and the presence of the fungi as colonizers.

Response to Reviewer 2's comment: We thank the reviewer for his/her positive comment and for the valuable feedback. The role of fungal colonizers and the immune status of the patients are two very important and inherently linked factors. As the dataset increases in population size, we will pay attention to these.

Reviewer 2 comment: I would suggest more emphasize on possible associations between different species of Candida isolated in the infections and their presence as colonizers, in the discussions.

Response to Reviewer 2 comment:  We thank the reviewer for the valuable input. Unfortunately, the small number of Candida CRBSI events, particularly of events caused by non-albicans isolates, totaling six events, doesn’t allow for detailed subgroup analyses, as noted in the study’s limitations. Furthermore, we did not collect data on the species of Candida spp. involved in the colonization.

After reviewing the data, we noted that all patients with C.tropicalis (patient number=1), C.kefyr (1), and C.glabrata (2) CRBSI episodes were colonized at one body site on average. However, no patient with C.parapsylosis (2) was colonized by Candida species. Our research group is broadly exploring the role of risk factors for CRBSI events, including colonizers and the immune system, and further work on immunocompromised patients is being planned. We will gladly share our findings with you when the time comes. Still, as of now, it would be imprecise and hasty for us to make further remarks on the role of fungal colonization in different non-albicans species considered singly.

Reviewer 2 comment: Some sentences need to be reformed and some typos to be corrected.

Response to Reviewer 2's comment: We thank the reviewer for his/her correction. We redacted the document, improved its fluency, and corrected typos.

Answers to Reviewer 3

Reviewer 3 comment: This study has very clear aims and these are very clearly laid out, but authors may want to comment on how applicable their findings will be to the rest of the world.

Response to Reviewer 3's comment: We thank the reviewer for the positive comments and constructive criticism. In the Aims of the Study section of the Background section (lines 84-88), a short sentence on our findings' applicability and future relevance has been added.

Reviewer 3 comment: The introduction is generally very well written, but contains a lot of acronyms and spends a lot of time discussing CABSI and CLBSI.  If the study is specifically going to analyse those with CRBSIs, the introduction could be modified to remove much of the discussion of CABSI and CLBSI, to ensure that the reader is aware of the what constitutes CRBSI and which patients would not be included and why.

Response to Reviewer 3's comment: We thank the reviewer for his/ her comment. The comments on the CABSI and CLABSI definitions have been summarized in the backgrounds section (lines 54-56), focusing on the CRBSI definition and why some clear CRBSI cases were not defined as such (lines 70-72).

Reviewer 3 comment: The results section needs some work to ensure that the results presented are relevant to the study question posed.  For instance, it is not necessary to know more than the number of patients with a catheter unless the authors are going to analyse differences between those that developed a CRBSI and those that did not.

Table 1 needs to be modified... the table clearly define the total number of patients and VAD placements, then include the number of CRBSI (with percentage), number of CABSI (with percentage), Candida species CRBSI ( with percentage) and Candida albicans CRBSI (with percentage). Incidence could also be included. Other items in the table are not required for the study unless you are comparing the parameters for those who got an infection with those who did not. Try to stick to the same number of decimal places.

Response to Reviewer 3's comment: We thank the reviewer for his/ her precise remarks on the first part of the results section. Population data, comorbidities in particular, have been removed from the description in the results sections and Table 1 as we did not intend to compare the small (17) number of patients with Candida CRBSI with more than a thousand patients without any VAD-related complication. Incidence values per 1000 catheter days were provided (Table 1), as well as for different VAD types, as discussed in the appropriate section of the paper. Decimal places were corrected accordingly in Tables 1,2 and 3.

Reviewer 3 comment: Figure 2 needs to use decimal points not commas and needs to italicise the organism names. Can include (n= ) for each species, as well as percentage.

Response to Reviewer 3's comment: we thank the reviewer for the remark, Figures 1 and 2 have been corrected accordingly

Reviewer 3 comment: line 193-197 is not really needed unless you are going to analyse the populations of those that got infections and that did not. Note that you cannot really say that it was primarily females, as it was just a little of 50%.

Response to Reviewer 3's comment: we thank the reviewer for his/her feedback, and this part was modified accordingly (line 157).

Reviewer 3 comment: Co-morbidities are more interesting if we are trying to link to likelihood of getting an infection. If you want to describe the study population, this would be more appropriate before describing the incidence of infections.

Response to Reviewer 3's comment: We thank the reviewer for the effort to improve our manuscript. The description of the study population is now brief (lines 154-162) and placed before the description of the infections’ prevalence and incidence rates

Reviewer 3 comment: lines 198-200 - only need to discuss the data relevant to the CRBSI... i.e. 16 of the 54 patients with CRBSI died (mortality of 29.6%), with 7 related to CRBSI. MORE important is how many were Candida CRBSI?

Response to Reviewer 3's comment: We thank the reviewer for the constructive criticism. The part related to Candida CRBSI mortality data is described in Table 3 and in the first part of the Results section paragraph (lines 180-181), following the study population’s mortality data.

Reviewer 3 comment: note sure that you need to include lines 201-208

Response to Reviewer 3's comment: we thank the reviewer for his/ her feedback; the results part has been re-structured according to his/her remarks

Reviewer 3 comment: Table 2 needs a title. For p values that reach significance, put these in bold to make them stand out. Increase the width of the first column to get each parameter on a single line can reduce the width of the other three columns.

Response to Reviewer 3's comment: Thanks for the precise suggestions. Table 2 has been formatted, providing a short title, the use of bold for statistically significant p-values, and improved use of the table space

Reviewer 3 comment: As the main aim of the study is to look at Candida CRBSI, I would move section 3.3 earlier in the results. Seems odd to refer to figure 1 so late in results. Some of the results mentioned are in Table 2 rather than Table 3. Some data are repeated in both tables.

Response to Reviewer 3's comment: We thank the reviewer for the valuable suggestion; we have moved Section 3.3 earlier (it is now called Section 3.2, and Table 3 is now called Table 2; as a result, former Table 2 and Section 3.2 are now named Table 3 and section 3.3, respectively) to follow up closely on the now brief description of the study population. Repetitions have been corrected accordingly.

Reviewer 3 comment: Reordering the results section to concentrate on the main focus of the study would be beneficial.  It would also have been nice to have analysed the C. albicans CRBSI vs non-albicans Candida CRBSI, which would have been 11 vs 6 infections - are there patient parameters that are linked to likelihood of developing a CRBSI?

Response to Reviewer 3's comment: We thank the reviewer for his/ her criticism; the results section has been redacted according to your suggestions. The new results section now features a brief description of the study population and CRBSI rates for candida and other bacteria, followed by a more thorough description of Candida spp. CRBSI events. Lastly, data on the comparison between Candida and bacterial CRBSI are displayed. We did not pursue further analyses comparing albicans and non-albicans Candida CRBSI because the numbers were exiguous. However, this topic is clearly of interest and will be dealt with in the future years as the dataset expands, including more fungal CRBSI episodes.

Reviewer 3 comment: This was a very interesting paper and very useful clinical information but will benefit from some additional work on the main text and the discussion could include a little more about the published literature regarding SARS-CoV-2 and Candida/fungal BSI.

Response to Reviewer 3's comment: To follow up on the reviewer’s precious insights, for which we are very thankful, we added a short, referenced paragraph on the impact of Candida spp. superinfections in COVID-19 patients during the pandemic (lines 289-296) to give greater importance to this very relevant and unanswered topic.

Reviewer 3 comment: consider comparing patient parameters for those who got a CRBSI with those that did not - which parameters differ, if any?

Response to Reviewer 3's comment: we thank the reviewer for his/her helpful comment; we did not compare patients’ parameters for those who did and did not have a CRBSI diagnosis as of now because of the great differences in the sample sizes. Furthermore, our team is already working on different methods to appropriately investigate comorbidities and other risk factors potentially involved in developing CRBSI. This is currently being done via the creation of a Machine Learning (ML) model and will be the object of a future publication in the year 2025 that we would be glad to share with the reviewer once ready. Lastly, we will also work on further comparisons between healthy and CRBSI patients using traditional statistics, which will provide insightful data once the dataset increases in sample size.

Reviewer 3 comment: modify Table 1

Response to Reviewer 3's comment: Table 1 has been modified according to the reviewer’s comments.

Reviewer 3 comment: move Candida data and information earlier

Response to Reviewer 3's comment: Candida's data have been moved to the top, turning section 3.3 into section 3.2 and table 3 into table 2; we agree that the results section is now more fluent with this change.

Reviewer 3 comment: add more in discussion about links between SARS-CoV-2 and Candida/fungal BSI

Response to Reviewer 3's comment: To follow up on the reviewer’s precious insights, we added a short, referenced paragraph on the impact of Candida spp. superinfections in COVID-19 patients during the pandemic to give greater importance to this very relevant and unanswered topic.

Reviewer 3 comment: Text modifications required:

Introduction

1)           line 60-61: "CABSI and CLABSI surveillance definitions may overestimate the true incidence of catheter-related BSIs (CRBSIs)." It is not clear why the authors believe this to be true.  Please clarify or cite supporting literature.

2)           line 74 " CRBSIs caused by Candida spp. have a significant mortality risk." reference?  Suggests that the bacterial ones do not - is this true?

3)           line 77 "Literature on CRBSIs involving CVADs has remained consistent in recent years," - what do that authors mean by this statement?  This is not clear.

4)           Line 81 "Moreover, despite Candida spp. remains a leading cause of CRBSIs," - take care with tense... should read "Moreover, despite Candida spp. remaining a leading cause of CRBSIs"

Response to Reviewer 3's comment: We thank the reviewer for his/ her precise comments on the text modifications required; all four points in the Introduction section have been addressed accordingly

Reviewer 3 comment: Materials and methods

line 92 - add inverted commas after SIP and before the citation

line 111 - candidiasis does not need a capital letter

line 112  - Invasion does not need  a capital letter (assuming that this is still part of the previous sentence.

line 121 - need citations for the two sources

line 122-123 - so are CLABSI a subset of CABSI?

line 127 - do both criteria need to be fulfilled for CRBSI?

Response to Reviewer 3's comment: We thank the reviewer for his/ her precise comments on text modifications required; all points concerning the materials and methods section have been addressed, regarding this comment in particular “line 122-123 - so are CLABSI a subset of CABSI?” our answer is yes, according to the most recent guidelines CABSI is now identified as a macro-category including all (peripheral and central) venous catheter infections for which diagnostic criteria for CRBSI were not pursued/ applicable.

Reviewer 3 comment: Results

line 175 - Candida spp. were involved, not was involved Reference to figure needs to be inserted before the full stop.  Use decimal points rather than commas.  Include in this paragraph that the majority of Candida CRBSI  (n=11) were caused by C. albicans.

Lines 177-179 need to be moved into the previous paragraph.

Line 180 and 181 - insert a space between S. and epidermidis. - do not need to define that Staphylococcus epidermidis will be shortened to S. epidermidis - this is a standard convention.

line 183 - remove text.

line 188-189 & 191- sensitive rather than sensible

line 243 - needs to start a new paragraph - line 247 would be part of the new paragraph.

Response to Reviewer 3's comment: we thank the reviewer for his/her precious feedback greatly improved the fluency of the results of the study section. The text has been corrected accordingly.

Reviewer 3 comment: Discussion

line 278 - what is that range?

line 290 - need to cite the reference number too

line 292-294  - has this been found previously?

line 314 is not a new paragraph - continue from previous line

lines 320-339 - paragraph structure needs reviewing

Response to Reviewer 3's comment: we thank the reviewer for the valuable comments regarding the discussion section of our study, these changes have been implemented accordingly.

Reviewer 2 Report

Comments and Suggestions for Authors

The subject is not new, but it brings more details about the association between fungal CRBSI and some data of the patients, such as their immune status and the presence of the fungi as colonizers.

I would suggest more emphasize on possible associations between different species of Candida isolated in the infections and their presence as colonizers, in the discussions.

Comments on the Quality of English Language

Some sentences need to be reformed and some typos to be corrected.

Author Response

(The authors gave the same response as above.)

Reviewer 3 Report

Comments and Suggestions for Authors

This study has very clear aims and these are very clearly laid out, but authors may want to comment on how applicable their findings will be to the rest of the world. 

The introduction is generally very well written, but contains a lot of acronyms and spends a lot of time discussing CABSI and CLBSI.  If the study is specifically going to analyse those with CRBSIs, the introduction could be modified to remove much of the discussion of CABSI and CLBSI, to ensure that the reader is aware of the what constitutes CRBSI and which patients would not be included and why.

The results section needs some work to ensure that the results presented are relevant to the study question posed.  For instance, it is not necessary to know more than the number of patients with a catheter unless the authors are going to analyse differences between those that developed a CRBSI and those that did not. 

Table 1 needs to be modified... the table clearly define the total number of patients and VAD placements, then include the number of CRBSI (with percentage), number of CABSI (with percentage), Candida species CRBSI ( with percentage) and Candida albicans CRBSI (with percentage). Incidence could also be included. Other items in the table are not required for the study unless you are comparing the parameters for those who got an infection with those who did not. Try to stick to the same number of decimal places.

Figure 1 needs to use decimal points not commas and needs to italicise the organism names. Can include (n= ) for each species, as well as percentage. Note that this figure does not show that 5 infections were were bacteria (not other bacteria as Candida spp are not bacteria).  Where is the co-infection data shown? 

Figure 2 needs to use decimal points not commas and needs to italicise the organism names. Can include (n= ) for each species, as well as percentage.

line 193-197 is not really needed unless you are going to analyse the populations of those that got infections and that did not. Note that you cannot really say that it was primarily females, as it was just a little of 50%. 

Co-morbidities are more interesting if we are trying to link to likelihood of getting an infection. If you want to describe the study population, this would be more appropriate before describing the incidence of infections.

lines 198-200 - only need to discuss the data relevant to the CRBSI... i.e. 16 of the 54 patients with CRBSI died (mortality of 29.6%), with 7 related to CRBSI. MORE important is how many were Candida CRBSI? 

note sure that you need to include lines 201-208

Table 2 needs a title. For p values that reach significance, put these in bold to make them stand out. Increase the width of the first column to get each parameter on a single line can reduce the width of the other three columns.

As the main aim of the study is to look at Candida CRBSI, I would move section 3.3 earlier in the results. Seems odd to refer to figure 1 so late in results. Some of the results mentioned are in Table 2 rather than Table 3. Some data are repeated in both tables. 

Reordering the results section to concentrate on the main focus of the study would be beneficial.  It would also have been nice to have analysed the C. albicans CRBSI vs non-albicans Candida CRBSI, which would have been 11 vs 6 infections - are there patient parameters that are linked to likelihood of developing a CRBSI? 

This was a very interesting paper and very useful clinical information, but will benefit from some additional work on the main text and the discussion could include a little more about the published literature regarding SARS-CoV-2 and Candida/fungal BSI. 

Main modifications:

1. consider comparing patient parameters for those who got a CRBSI with those that did not - which parameters differ, if any? 

2. modify Table 1 

3. move Candida data and information earlier 

4. add more in discussion about links between SARS-CoV-2 and Candida/fungal BSI

Text modifications required:

Introduction - line 60-61: "CABSI and CLABSI surveillance definitions may overestimate the true incidence of catheter-related BSIs (CRBSIs)." It is not clear why the authors believe this to be true.  Please clarify or cite supporting literature. 

line 74 " CRBSIs caused by Candida spp. have a significant mortality risk." reference?  Suggests that the bacterial ones do not - is this true? 

line 77 "Literature on CRBSIs involving CVADs has remained consistent in recent years," - what do that authors mean by this statement?  This is not clear. 

Line 81 "Moreover, despite Candida spp. remains a leading cause of CRBSIs," - take care with tense... should read "Moreover, despite Candida spp. remaining a leading cause of CRBSIs"

Materials and methods

line 92 - add inverted commas after SIP and before the citation

line 111 - candidiasis does not need a capital letter

line 112  - Invasion does not need  a capital letter (assuming that this is still part of the previous sentence.

line 121 - need citations for the two sources 

line 122-123 - so are CLABSI a subset of CABSI? 

line 127 - do both criteria need to be fulfilled for CRBSI?

Results

line 175 - Candida spp. were involved, not was involved Reference to figure needs to be inserted before the full stop.  Use decimal points rather than commas.  Include in this paragraph that the majority of Candida CRBSI  (n=11) were caused by C. albicans

Lines 177-179 need to be moved into the previous paragraph. 

Line 180 and 181 - insert a space between S. and epidermidis. - do not need to define that Staphylococcus epidermidis will be shortened to S. epidermidis - this is a standard convention. 

line 183 - remove text. 

line 188-189 & 191- sensitive rather than sensible

line 243 - needs to start a new paragraph - line 247 would be part of the new paragraph. 

Discussion 

line 278 - what is that range? 

line 290 - need to cite the reference number too

line 292-294  - has this been found previously? 

line 314 is not a new paragraph - continue from previous line

lines 320-339 - paragraph structure needs reviewing 

Comments on the Quality of English Language

The quality of the English language is very good but there are some issues with paragraph structure, particularly in the discussion.  This is easily dealt with. 

Author Response

(The authors gave the same response as above.)

Round 2

Reviewer 1 Report

Comments and Suggestions for Authors

Dear authors I apologize for sending you a wrong document. Unfortunately I did not have the chance to send the proper one.

I am now sending you separately my comments on the new one.

Comments on the Quality of English Language

English still need enough corrections.

Author Response

Dear Editor,

Enclosed is our revised manuscript entitled “Epidemiology and Clinical Insights of Catheter-related Can-didemia in Non-ICU Patients with Vascular Access Devices.” We want to thank the Editor and the reviewers for their constructive criticism, which substantially improved our paper. We hope that the paper will be suitable for publication. We have made a point-by-point answer to the referees’ comments below.

Answers to Reviewer 1

Reviewer 1 comment:  

  1. Line 72. Surveillance terms. Are those surveillance terms explained somewhere in the text?

Response to Reviewer 1 comment: We thank the reviewer for his/her question. As specified in the first revised version, the definitions are according to Infusion Therapy Standards of Practice, 9th Edition, published in January 2024, quote number 6. Moreover, the definitions are reported in the methods section, lines 118-132. We removed the definition in the first revised version and left only the reference to satisfy the reviewer's three suggestions. Now, we have reported the definition in the text.

Reviewer 1 comment: 

  1. Line 77. Remaining “. Remain should be the word

Response to Reviewer 1 comment: We thank the reviewer for his/her correction. We have changed the test as suggested.

Reviewer 1 comment: 

  1. Line 83. “ Risk information “ makes no sense

Response to Reviewer 1 comment: We thank the reviewer for his/her correction. We have changed the test. “This data on Candida CRBSI microbiological features and patients’ characteristics could aid clinicians in identifying at-risk individuals who may benefit from empirical antifungal therapies in non-intensive care settings.”

Reviewer 1 comment: 

  1. Line 109 to 112. one understands by this sentence that candidemia included both positive BC and an organ candidiasis. Is that what you mean?

Response to Reviewer 1's comment: We thank the reviewer for his/her correction. We have changed the paragraph to refer to confirmed deep-seated candidiasis, which is defined as organ candidiasis and positive BC for Candida.

Reviewer 1 comment: 

  1. Line 134. CABSI/CLABSI. Is there a definition anywhere in the text ? Furthermore do you mean CA-BSI OR CABSI according to the nurse definition?

Response to Reviewer 1 comment: We thank the reviewer for his/her question. As specified in the first revised version, the definitions are according to Infusion Therapy Standards of Practice, 9th Edition, published in January 2024, quote number 6. Moreover, the definitions are reported in the methods section, lines 118-132. We removed the definition in the first revised version and left only the reference to satisfy the reviewer's three suggestions. Now, we have reported the definition in the text.

To facilitate the reviewer work, there is a brief paragraph of the INS 2024: “Given variability in international definitions, outcome reporting, and application of the terms catheter-related bloodstream infection (CR-BSI) and central line-associated bloodstream infection (CLABSI), the INS Standards of Practice Committee is using the terminology Catheter-Associated Bloodstream Infection (CABSI) to refer to bloodstream infections (BSIs) originating from either peripheral intravenous catheters (PIVCs) and/or central vascular access devices (CVADs). Both are equally injurious...”.

The correct form is CABSI.

The Infusion Therapy Standards of Practice is a bible for all those who work in the vascular access world (nurses and physicians) and is quoted in all papers about vascular access. Even if this document written by nurses has some criticisms, this paper is the only one that evaluates the problem of vascular access infection diagnosis every three years.

The latest “guidelines” performed by physicians are the Spanish guidelines of 2018 “Chaves F, Garnacho-Montero J, Del Pozo JL, Bouza E, Capdevila JA, de Cueto M, Domínguez MÁ, Esteban J, Fernández-Hidalgo N, Fernández Sampedro M, Fortún J, Guembe M, Lorente L, Paño JR, Ramírez P, Salavert M, Sánchez M, Vallés J. Diagnosis and treatment of catheter-related bloodstream infection: Clinical guidelines of the Spanish Society of Infectious Diseases and Clinical Microbiology and (SEIMC) and the Spanish Society of Spanish Society of Intensive and Critical Care Medicine and Coronary Units (SEMICYUC). Med Intensiva (Engl Ed). 2018 Jan-Feb;42(1):5-36. English, Spanish. doi: 10.1016/j.medin.2017.09.012. PMID: 29406956.”

I suggest the reviewer have a little more respect for “nurse” work because, in some matters, they are better than physicians, and vascular access is one such matter.

Reviewer 1 comment: 

  1. Line 162. “Midline catheters were the most frequent VADs” you mean frequently placed

Response to Reviewer 1's comment: We thank the reviewer for his/her correction and have changed the test as suggested.

Reviewer 1 comment: 

  1. Line 222. “Skin levels “ not a proper expression

Response to Reviewer 1 comment: We thank the reviewer for his/her correction. We have changed the test as suggested.

Reviewer 1 comment: 

  1. Line 271 being diagnosed with Candida spp. CRBSIs were” : you should express it in another way

Response to Reviewer 1's comment: We thank the reviewer for his/her correction and have changed the test as suggested.

Reviewer 1 comment: 

  1. Line 281. Dwelling times, not dwell

Response to Reviewer 1's comment: We thank the reviewer for his/her correction and have changed the test as suggested.

Reviewer 1 comment: 

  1. Line 311. CRBSI, scarce” look again at those two words. Perhaps you should use a full stop and not a coma

Response to Reviewer 1 comment: We thank the reviewer for his/her correction. We have changed the test as suggested.

Reviewer 1 comment: 

  1. Line 319. Candida spp. CRBSI patients : please express it in another way

Response to Reviewer 1 comment: We thank the reviewer for his/her correction. We have changed the test as suggested.

Reviewer 1 comment: 

  1. Lines 326 – 328 this does not make sense

Response to Reviewer 1 comment: We thank the reviewer for his/her correction. We have removed the sentence as suggested.

Reviewer 1 comment: 

Line 331. Despite non-albicans, Candida : please see if a verb is missing

Response to Reviewer 1 comment: We thank the reviewer for his/her correction. We have changed the test as suggested.

Reviewer 1 comment: 

Line 340. increases its significance by : the Covid 19 timeframe increases the significance of the study?

Response to Reviewer 1 comment: We thank the reviewer for his/her correction. We changed the sentence as follows: “Moreover, the study’s timeframe included the COVID-19 pandemic, which undoubtedly increased the number of catheter-related infections outside the ICU [23].”

Reviewer 1 comment: 

Lines 345 – 346. there is a limited number of Candida spp. Infections did not allow for extensive subgroup analyses : look again at these sentences . Is a full stop or coma needed? What do you mean by them?

Response to Reviewer 1 comment: We thank the reviewer for his/her correction. We have changed the test as suggested.

Reviewer 1 comment: 

Why do you report the overall CABSI prevalence in the table 1 since you do not comment it in the discussion?

Response to Reviewer 1 comment: We thank the reviewer for his/her question. We kept the CABSI data in the table to show the reader how precise our hospital was in pursuing appropriate diagnostics for catheter-related infections. In all our preceding works, we always presented CRBSI data followed by CABSI or CLABSI rates as “quality controls” of our CRBSI estimates.